# Topological Fermi-arc surface state covered by floating electrons on a two-dimensional electride

Chan-young Lim [1,10,11], Min-Seok Kim [2,11], Dong Cheol Lim[3,4,11], Sunghun Kim [5,11], Yeonghoon Lee[6], Jaehoon Cha[1], Gyubin Lee[1], Sang Yong Song [2], Dinesh Thapa[7], Jonathan D. Denlinger [8], Seong-Gon Kim [9] ✉, Sung Wng Kim [3,4] ✉, Jungpil Seo [2] ✉ & Yeongkwan Kim [1] ✉

Two-dimensional electrides can acquire topologically non-trivial phases due to intriguing interplay between the cationic atomic layers and anionic electron layers. However, experimental evidence of topological surface states has yet to be verified. Here, via angle-resolved photoemission spectroscopy (ARPES) and scanning tunnelling microscopy (STM), we probe the magnetic Weyl states of the ferromagnetic electride $[Gd_2C]^{2+} \cdot 2e^-$. In particular, the presence of Weyl cones and Fermi-arc states is demonstrated through photon energy-dependent ARPES measurements, agreeing with theoretical band structure calculations. Notably, the STM measurements reveal that the Fermi-arc states exist underneath a floating quantum electron liquid on the top Gd layer, forming double-stacked surface states in a heterostructure. Our work thus not only unveils the non-trivial topology of the $[Gd_2C]^{2+} \cdot 2e^-$ electride but also realizes a surface heterostructure that can host phenomena distinct from the bulk.

Electrides are ionic crystals where electrons play the role of anions, squeezed between the positively charged atomic layers. Further, the symmetry of two-dimensional (2D) electrides, such as $[Gd_2C]^{2+} \cdot 2e^-$ and $[Y_2C]^{2+} \cdot 2e^-$, imposes the topological texture on their bulk electronic structure[1-3]. Under the unique electrostatic and topological properties, 2D electrides hold a compelling potential to host different kinds of surface states at the edges of the system[1-12]. That is, floating electrons are confined at the vacuum side over the top terminated surface, compensating for the positive charge of the exposed surface

layer[4-10]. In addition to these characteristic surface states, topological surface states can also be located at the uppermost atomic layer[1-3,11-13]. Specifically, $[Gd_2C]^{2+} \cdot 2e^-$ and $[Y_2C]^{2+} \cdot 2e^-$ are predicted to host Fermi arc and drumhead surface states[2,3], as a counterpart of Weyl and Dirac fermions in bulk. Consequently, the floating electron and topological surface states assemble into an intriguing double-stacked heterostructure.

Although the floating electrons on the surface of $[Gd_2C]^{2+} \cdot 2e^-$ were experimentally captured with sizable electron correlation strength

[1]Department of Physics, Korea Advanced Institute of Science and Technology, Daejeon 34141, South Korea. [2]Department of Physics and Chemistry, Daegu Gyeongbuk Institute of Science and Technology, Daegu 42988, South Korea. [3]Department of Energy Science, Sungkyunkwan University, Suwon 16419, South Korea. [4]Center for Electride Materials, Sungkyunkwan University, Suwon 16419, South Korea. [5]Department of Physics, Ajou University, Suwon 16499, South Korea. [6]Quantum Spin Team, Korea Research Institute of Standards and Science, Daejeon 34113, South Korea. [7]Department of Chemistry and Biochemistry, North Dakota State University, Fargo, ND 58108, USA. [8]Advanced Light Source, Lawrence Berkeley National Laboratory, Berkeley, CA 94720, USA. [9]Department of Physics & Astronomy and Center for Computational Sciences, Mississippi State University, Mississippi State, MS 39792, USA. [10]Present address: Donostia International Physics Center (DIPC), 20018 San Sebastián/Donostia, Spain. [11]These authors contributed equally: Chan-young Lim, Min-Seok Kim, Dong Cheol Lim, Sunghun Kim. ✉e-mail: sk162@msstate.edu; kimsungwng@skku.edu; jseo@dgist.ac.kr; yeongkwan@kaist.ac.kr

evidenced by its liquid nature[9], the existence of topological surface states has not been demonstrated experimentally. To complete the presence of these unique surface states at the edge of 2D electrides, together with unique stacking formation, we investigated the electronic structure of $[Gd_2C]^{2+}\cdot2e^-$ using angle-resolved photoemission spectroscopy (ARPES), scanning tunneling microscopy (STM) and density functional theory (DFT) calculations, focusing on the topological aspects of Weyl fermions in the bulk states and Fermi-arc states at the surface.

## Results

To understand the non-trivial surface states of $[Gd_2C]^{2+}\cdot2e^-$, we studied the band structure of $[Gd_2C]^{2+}\cdot2e^-$ with DFT slab calculations. Figure 1b−d show the calculated band contribution of surface floating electrons, topmost Gd atomic orbitals, and interstitial anionic electrons (IAEs) localized between the layers along the high symmetry directions, respectively. Red and blue colors represent spin up and down states, split by the bulk ferromagnetic order of $[Gd_2C]^{2+}\cdot2e^{-\ 14,15}$. It is noteworthy that the bands with dominant contribution from the electrons in the topmost Gd atoms are distinct from previously observed floating bands and bulk IAE bands[8], implying the existence of additional surface states at the topmost atomic layer. In detail, the pair of parabolic bands along $\bar{\Gamma} - \bar{K}$ point ascribes to spin up/down bands of floating electrons. Distinguished from the floating electrons, the additional surface character is captured for the band along $\bar{K} - \bar{M}$ direction (purple arrow in Fig. 1c), which follows the predicted dispersion of Fermi-arc states in the previous report[3].

Figure 1e displays a Fermi surface (FS) of $[Gd_2C]^{2+}\cdot2e^-$ measured by ARPES. Apparently, all the band structures of IAEs, floating electrons, and Fermi arc are captured in the FS, indicating that they are located within the probing depth of ARPES. The circular FS centered at $\bar{\Gamma}$ point originates from the floating electrons[9], and their Fermi liquid-like dispersion is confirmed in Fig. 1f. The complex FS around $\bar{\Gamma}$ point is mostly from $[Gd_2C]^{2+}$ layers and IAEs. Noticeably, the clue of the predicted Fermi-arc states is found near $\bar{K}$ points, as marked by the dashed curves in Fig. 1e, which will be discussed in detail.

Among the key band features measured by ARPES, we first investigate the floating electrons in real space using STM. Figure 2a shows an STM image measured on the as-cleaved $[Gd_2C]^{2+}\cdot2e^-$ sample,

capturing the floating electrons on the surface, as the nearest states to the STM tip. Figure 2b depicts the height profile obtained along the dotted line in Fig. 2a. The step height is approximately 6 Å, agreeing with the atomic structure of $[Gd_2C]^{2+}\cdot2e^-$ (ref. 14). The ARPES measurement reveals that the band onset of the floating electrons lies 0.25 eV below the Fermi energy (the inset in Fig. 2c), which is represented as a peak in the STM spectrum (Fig. 2c). Owing to the delocalized nature of the floating electrons, quasiparticle interference (QPI) patterns are expected to be observed around the impurities and step edges. Figure 2d, e depict an STM image obtained near the Fermi energy (−50 meV) and its differential conductance ($dI/dV$) map, respectively. The QPI patterns in the $dI/dV$ map are isotropic, and their Fourier transformation (FT) shows that the modulation vectors ($\mathbf{q}$s) form a ring shape around the $\bar{\Gamma}$ point (Fig. 2h). The size of the modulation vectors agrees well with the scattering vectors taken within the surface electron pocket measured by ARPES, as indicated by the red arrows in Fig. 2g, h. As the STM bias voltage is decreased, the size of $\mathbf{q}$ decreases, resulting in a longer wavelength in the QPI pattern (Fig. 2f, i). The decreasing trend of q by the bias voltage aligns with the scattering vectors extracted from the surface electron pocket measured by ARPES (Fig. 2j and Supplementary Fig. 1).

Turning our focus to the band topology of $[Gd_2C]^{2+}\cdot2e^-$, Weyl points (WP) accessible with ARPES ($W_{13}$ and $W_{14}$ points) are investigated. They are expected to be located near $k_z = \pi$ plane along $\bar{\Gamma} - \bar{M}$ high symmetry line as represented with blue and red spheres in Fig. 3a. The dispersion measured along $\bar{\Gamma} - \bar{M}$ direction at $k_z = \pi$ plane is given in Fig. 3c. In the red box, the cone-like feature is captured, which is spreading from $W_{13}$ or $W_{14}$ points in the momentum space. Since Weyl cones have 3D dispersions, i.e., gapless only at the WP, the $k_z$ modulation of band dispersions should be examined. The upper panels of Fig. 3d are band dispersions along $\bar{\Gamma} - \bar{M}$ direction at different $k_z$ positions, and the lower panels are 2D curvature plots of the upper panels[16]. In both plots, the cone-like dispersions are clearly shown, although the intensity of one band is dominating among the two intersecting bands. The uneven intensity observed between two intersecting bands is frequently observed in cases of Dirac/Weyl crossings, resulting from the differing symmetries of the corresponding wave functions of each band concerning the experimental geometry of light polarization and incident light direction[17−20].

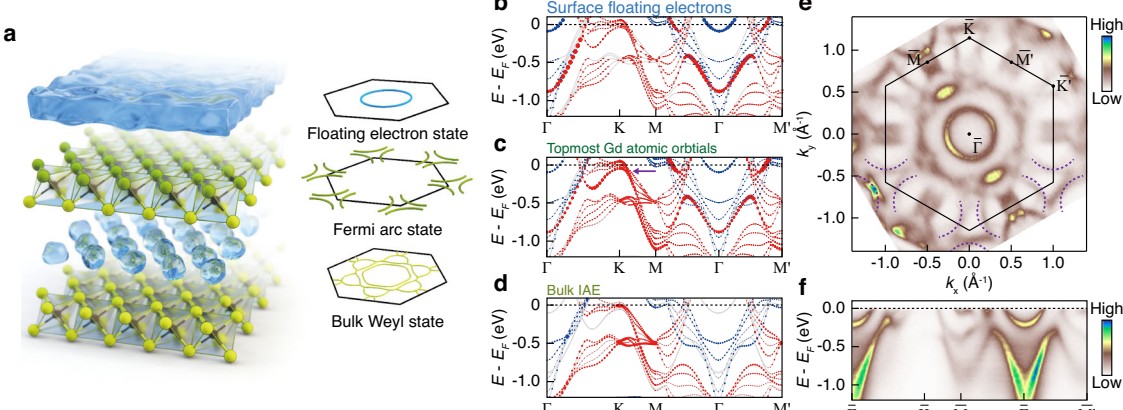

**Fig. 1 | Crystal structure and electronic structure of $[Gd_2C]^{2+}\cdot2e^-$. a** Schematic of the crystal (left panel) and electronic (right panel) structures. In the left panel, the green and black spheres represent Gd and C atoms, respectively, forming $[Gd_2C]^{2+}$ layers. Blue blobs between the $[Gd_2C]^{2+}$ layers denote bulk interstitial anionic electrons (IAEs). The floating electrons are depicted atop the crystal structure. The right panel illustrates the electronic structure corresponding to the crystal structure. The bulk of the crystal exhibits a non-trivial band topology, hosting the bulk Weyl state. The Fermi-arc state is observed on the $[Gd_2C]^{2+}$ surface, while the floating electron state reveals a

circular Fermi surface. **b–d** Calculated electronic structures with contributions from electrons at the surface floating electrons (**b**), the topmost Gd atoms (**c**), and the bulk IAEs (**d**). Blue and red colors denote major and minor spin components, respectively. The purple arrow in (**c**) marks the predicted Fermi-arc dispersion. **e, f** Fermi surface (**e**) and high-symmetric line dispersions (**f**) obtained from ARPES measurements with 90 eV photons. The Fermi surface was symmetrized along $\bar{\Gamma} - \bar{K}\prime$ direction. The black solid line on (**e**) represents the 1st Brillouin zone (BZ) boundary, and the purple dashed lines near $\bar{K}$ and $\bar{K}\prime$ points guide the expected Fermi arcs.

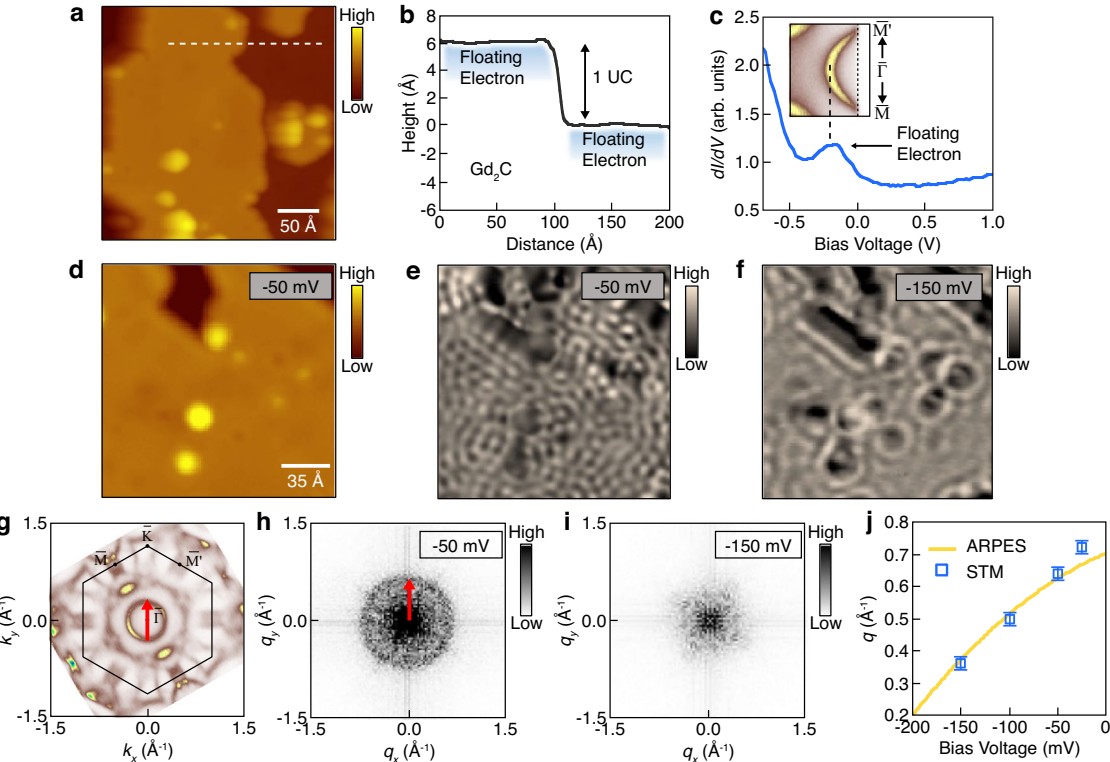

**Fig. 2 | STM measurements on floating electrons of [Gd₂C]²⁺·2e⁻. a** Topographic image of as-cleaved $[Gd_2C]^{2+}\cdot 2e^-$ sample. $V_{bias} = -100$ mV and $I_t = 100$ pA. **b** Height profile obtained along the dashed line in (**a**). There is one unit cell (UC) height difference between the neighboring terraces. **c** $dI/dV$ spectrum measured on the terrace of (**a**). The peak around $-0.25$ V indicates the onset of the band of floating electrons. The inset shows the band of floating electrons measured by ARPES. **d** Enlarged topographic image. $V_{bias} = -50$ mV and $I_t = 100$ pA. **e** $dI/dV$ map at $V_{bias} = -50$ mV. The quasiparticle interference (QPI) patterns are observed. Lock-in modulation amplitude ($V_{mod}$) of 10 mV is used. **f** $dI/dV$ map at $V_{bias} = -150$ mV. $V_{mod} = 10$ mV. **g** Fermi surface measured by ARPES. The red arrow indicates the nesting vector of the band of floating electrons. **h** Fourier transformed the image of (**e**). The QPI modulation vector (**q**) corresponds to the nesting vector in (**g**), which is marked by the same-sized red arrow. **i** Fourier transformed image of (**f**). **j** Dispersion of q by the bias voltage. The error bar represents the momentum uncertainty resulting from the resolution of $dI/dV$ maps.

The smooth intensity profile, lacking a true dip along the dominant band, can be attributed to the band crossing occurring without hybridization with the minor band. Hybridization typically encodes the character of the minor band into the dominant band. By considering the intensity profile of the dominant band, and by tracing the dispersion of the minor band against the dominant band, the intersecting point for $W_{13/14}^+$ was determined to be located at $k_z = \pi$, as predicted. This proves that the cone-shaped features are Weyl cones of $[Gd_2C]^{2+}\cdot 2e^-$.

Next, the photon energy dependence of the FS near $\bar{K}$ point was diagnosed to capture the signature of Fermi-arc states. While keeping the in-plane momentum window as in the top plot, FSs are accumulated with different photon energies ranging from 90 eV to 110 eV which covers half of the Brillouin zone (BZ) along $k_z$ direction from 0 to $\pi$ (Fig. 4). The schematics of the expected Fermi-arc states near the $\bar{K}$ points are overlaid on the measured FSs. Blue and red dots are WPs with opposite chirality, and black dashed line segments correspond to the Fermi-arc states. Left and right panels in Fig. 4 are raw images and their curvature plots, respectively. Remarkably, the shape of segments tightly follows the Fermi-arc states that are predicted in the calculations[3]. Furthermore, the segments of FS indicated with dashed lines are found to be intact across the photon energy. Compared to the apparent photon energy dependence of the adjacent bulk bands, this negligible photon energy dependence further shows the 2D nature of the Fermi-arc states. Therefore, we conclude that Fermi-arc states exist in the $[Gd_2C]^{2+}\cdot 2e^-$, which completes the presence of Weyl fermions, together with the captured Weyl cones.

As the last step, the Fermi-arc states are investigated in real space by STM. Since the floating electrons cover the surface of Gd₂C sample (Fig. 2a), STM cannot directly access the topmost Gd layer which harbors the Fermi arc. To remove such floating electrons, the sample was heated to 80 K and cooled to 4 K after an hour. Figure 5a shows the STM image of the $[Gd_2C]^{2+}\cdot 2e^-$ sample after the heating cycle. Remarkably, the floating electrons have vanished on the surface, exposing the Gd layer in the $[Gd_2C]^{2+}$. Only some amounts of floating electron residue remain, primarily near the step edges. Figure 5b illustrates the height profile taken along the dotted line in Fig. 5a. The 3 Å height of floating electrons is consistent with the theoretical calculation[8]. The removal of floating electrons is confirmed by the ARPES measurements, wherein the band of floating electrons is absent in the sample subjected to the heating cycle (inset of Fig. 5c and Supplementary Fig. 3). Accordingly, the floating electron peak is also missing in the STM spectrum (Fig. 5c).

Given that no floating electrons are present on the surface, the electrons under the surface must be accommodated to satisfy the electrostatic conditions. Within $[Gd_2C]^{2+}\cdot 2e^-$, IAEs are squeezed between $[Gd_2C]^{2+}$ layers, and their electron densities will change to screen the surface charges. Nevertheless, the ARPES data measured on Gd₂C, which underwent the same heating process as in STM, clearly show the presence of Fermi-arc states on the surface (Supplementary Fig. 3), although there is a slight Fermi level shift. This demonstrates that the charge redistribution does not alter the bulk topology of the system.

Figure 4d shows the larger area of the Gd surface measured at $V_{bias} = 25$ mV. The $dI/dV$ map obtained simultaneously with Fig. 5d is

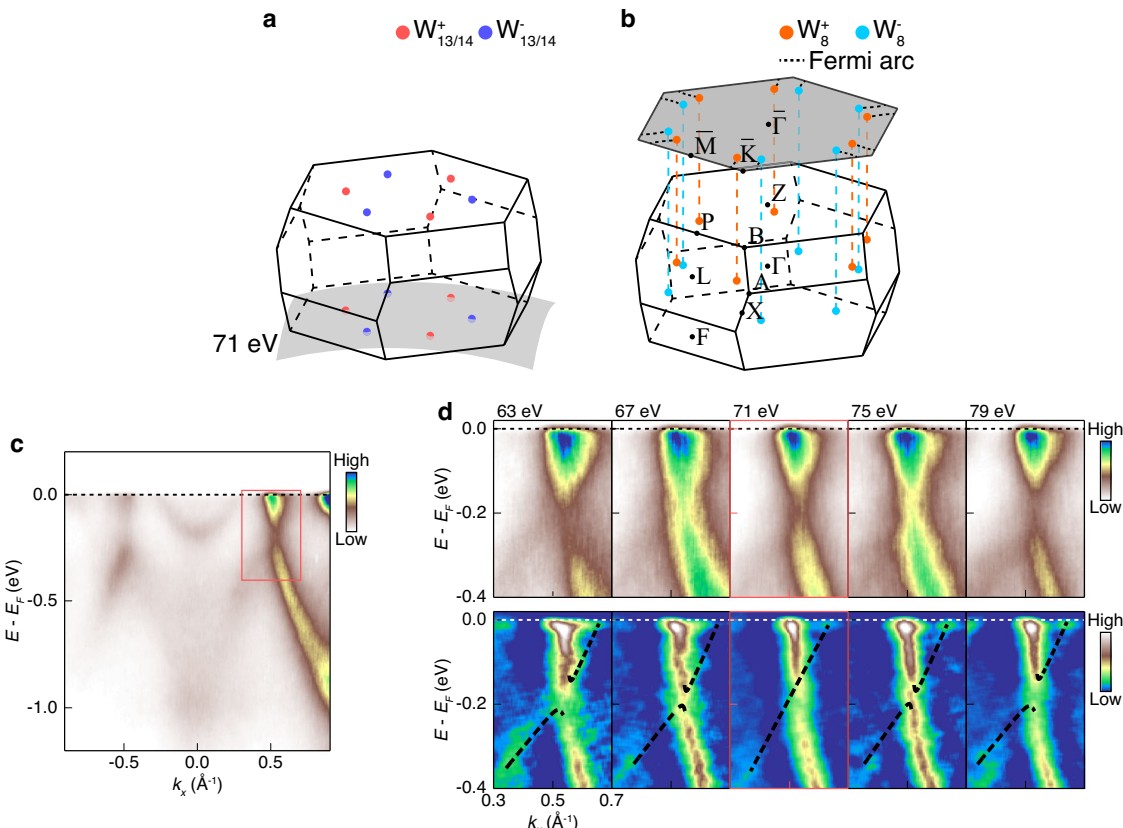

**Fig. 3 | ARPES measurements on Weyl cones in [Gd$_2$C]$^{2+}$·2e$^-$. a, b** Schematic 3D BZs of [Gd$_2$C]$^{2+}$·2e$^-$ displaying W$_{13/14}$ (**a**) and W$_8$ Weyl points (WPs) (**b**). Spheres with different colors in (**a**, **b**) represent WPs with opposite chirality. The gray surfaces **a**, **b** denote 71 eV ARPES measurement plane and the (111) surface-projected BZ, respectively. On the surface-projected BZ, WPs are connected to other WPs with opposite chirality at neighboring BZs via Fermi arcs drawn with black dashed line segments. **c** 71 eV $\bar{\Gamma} - \bar{M}$ dispersion with Weyl state inside the red box indicating expected Weyl cone position. **d** Weyl state measured with various photon energies near 71 eV. The upper panels are raw images, and the lower panels are their 2D curvature plots to enhance band features. Black dashed lines in the lower panels serve as guides for eyes to visualize the Weyl cones.

presented in Fig. 5e, where plentiful QPI patterns are developed. To understand the QPI patterns alongside the Fermi-arc states, we conducted FT analysis on the STM image and its d*I*/d*V* map. In the FT of the STM image (Fig. 6a), the lattice peaks (q$_{Bragg}$) of the Gd layer are identified, helping to define the BZ. The presence of the Gd lattice peaks indicates that there is no surface modification upon the removal of the floating electrons. Figure 6b displays the FT of the d*I*/d*V* map, revealing two distinct modulation vectors, q$_1$ along the $\bar{\Gamma} - \bar{M}$ direction and q$_2$ along the $\bar{\Gamma} - \bar{K}$ direction. The magnitude of q$_1$ extends up to 1.3 Å$^{-1}$ beyond the BZ, while q$_2$, slightly smaller than q$_1$, also reaches the BZ boundary. These large modulation vectors can be constructed by the nesting of scattering wavevectors between the Fermi-arc states positioned at the $\bar{K}$ and $\bar{K}\prime$ points, represented by the red and green arrows for q$_1$ and q$_2$ in Fig. 6d, respectively. The depiction of the Fermi-arc states in Fig. 6d is reproduced based on the ARPES data.

To further understand the nesting condition of the scattering wavevectors, we investigate the Fermi-arc states associated with the q$_1$ and q$_2$ vectors. In Fig. 6e, the q$_1$ vector (red arrow) joins the parallel segments of Fermi-arc states, highlighted by the dotted lines, across the BZ. Figure 6f shows that the q$_2$ vector (green arrow) connects the parallel segments of Fermi-arc states within the BZ. The spin degeneracy of the Fermi arc is already lifted by the bulk ferromagnetism and thus does not significantly affect the nesting conditions of q$_1$ and q$_2$[21], although the spin-momentum locking property imposes an additional helical spin texture on the Fermi-arc states[3,22]. Notably, the parallel segments associated with the q$_1$ and q$_2$ vectors are closely tied to the WPs that are fixed in the momentum space. Therefore, simply

speaking, the q$_1$ and q$_2$ vectors are linking the WPs. Altering the STM bias voltage could modify the detailed structure of the Fermi arc. However, as long as the WPs maintain their positions as topological objects, the sizes of q$_1$ and q$_2$ can remain constant by the bias voltage.

To check whether the q$_1$ and q$_2$ vectors are affected by the bias voltage, we lowered the bias voltage to −100 mV and obtained the QPI patterns (Fig. 5f). The FT of the QPI patterns is displayed in Fig. 6c. When compared to Fig. 6b, the positions of q$_1$ and q$_2$ remain relatively unaltered, although their intensities diminish. Figure 6g, h demonstrates that the nesting conditions for q$_1$ and q$_2$ can be loosened by lowering the bias voltage. However, the magnitudes of q$_1$ and q$_2$ do not change significantly since the Fermi-arcs states are anchored to the Weyl cone. This is further seen in the ARPES data, where the Fermi-arc states are mildly dispersive near the Weyl cone and thus the nesting vectors maintain their magnitudes (Supplementary Fig. 4). The overall trend of q$_1$ and q$_2$, depending on the bias voltage, is illustrated in Fig. 6i, confirming that the modulation vectors q$_1$ and q$_2$ are only weakly affected by the minimal changes in the Fermi-arc states near the Weyl cone. This robust dispersion of QPI modulation vectors, which is different from the dispersion of the Fermi liquid shown in Fig. 2j, again supports the topological nature of the Fermi-arc states of [Gd$_2$C]$^{2+}$·2e$^-$.

## Discussion

Present results not only confirm the existence of Weyl fermions and Fermi-arc states in [Gd$_2$C]$^{2+}$·2e$^-$ but also complete the full

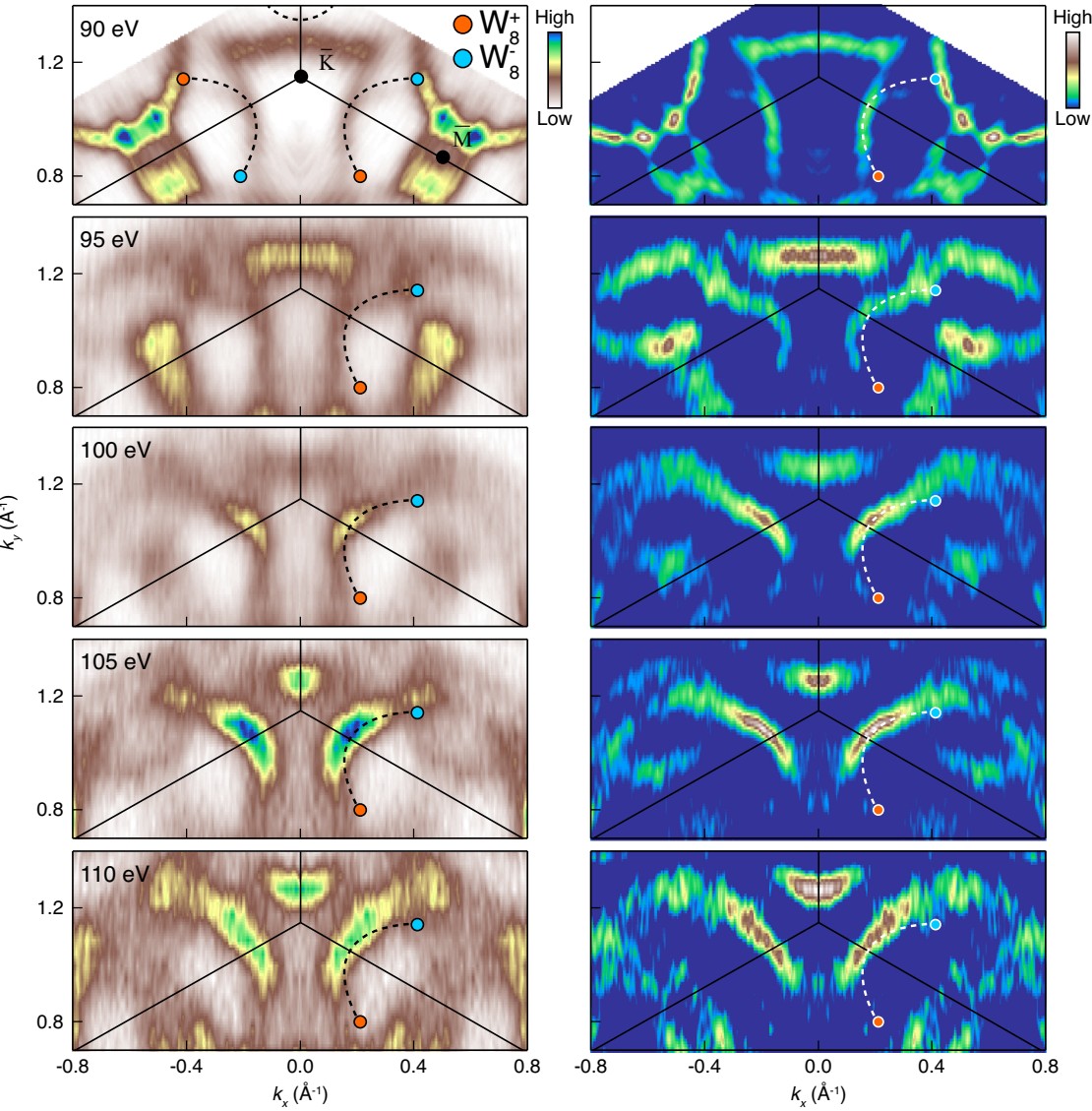

**Fig. 4 | ARPES measurements on Fermi arcs in [Gd₂C]²⁺·2e⁻.** Photon energy-dependent Fermi surfaces (FSs) near the K̄ point (left) and their 2D curvature plots to enhance FS features (right). Both panels are symmetrized with respect to $k_x = 0$ for better visualization of Fermi arcs. Light red and light blue points indicate $W_8^+$ and $W_8^-$ WPs, respectively. Black solid lines denote BZ boundaries. Fermi arcs are drawn with dashed line segments connecting $W_8$ WPs. In the original data for the 90 eV Fermi surface, all Fermi arcs are displayed, but they are omitted in the other plots except one used for comparison.

characterization of possible states at the edge of the system – a realization of heterosurface states with correlated electron liquid states and topological Fermi-arc states, as schematically illustrated in Fig. 1a. This resembles recent approaches of material engineering, combining correlated and topological materials into 2D heterostructure in order to add up the topological flavor to the correlated phenomena or vice versa. An example would be attaching superconductivity to topological materials to induce the topological superconducting state by injecting the cooper pairs into the topological state through proximity coupling[23–29]. Beyond that, this platform of double-stacked heterosurface states can lead to an emergent phenomenon that is hardly achieved in bulk heterostructures, by virtue of the unique characteristics of states at the surface. For instance, the non-trivial momentum dependence of Fermi-arc states, i.e., only the segment of FS exists, could provoke a non-trivial proximity effect that is limited for the electron within a specific momentum range where the Fermi arc exists, which has not been accounted for before. Therefore, the insights gained in this study will trigger to see how the two essential physical ingredients of non-trivial topology and pure electron-correlated

systems mutually influence each other in exploring emergent quantum phenomena[30].

## Methods

### Single crystal growth

All single crystal growth processes were carried out in high-purity Ar gas (99.999%). The floating zone melting method was used to grow single crystals of [Gd₂C]²⁺·2e⁻ as described in ref. 14. Polycrystalline [Gd₂C]²⁺·2e⁻ rods synthesized by the arc melting method were used as the feed and seed materials. The Gd metal and graphite pieces with molar ratio Gd: C = 2: 1 were mixed in the Ar-filled glove box, and then melted in an arc furnace. The melting process was repeated at least three times to achieve high homogeneity. The polycrystalline [Gd₂C]²⁺·2e⁻ was formed into rods after the melting and cooling processes. The feed and seed rods were rotated in opposite directions at the same speed of 6 rpm. The growth speed was slower than 2 mm per hour due to the low melt viscosity of the [Gd₂C]²⁺·2e⁻. The quality of the grown single crystal samples was checked using X-ray diffraction and inductively coupled plasma spectroscopy.

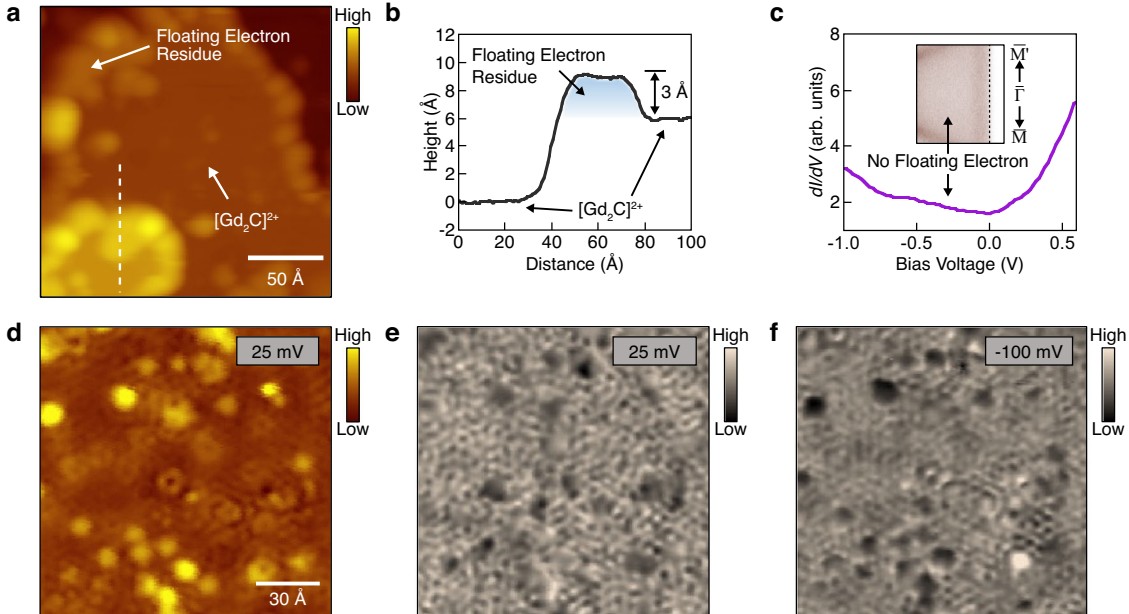

**Fig. 5 | STM measurements on Fermi-arc states of $[Gd_2C]^{2+}\cdot2e^-$. a** Topographic image of $[Gd_2C]^{2+}$ surface, where the floating electrons are removed by heating the sample at 80 K for 1 h. $V_{bias} = -100$ mV and $I_t = 100$ pA. **b** Height profile taken along the dashed line in (**a**). **c** $dI/dV$ spectrum measured on the $[Gd_2C]^{2+}$ surface. The band onset of the floating electrons is missing in the $dI/dV$ spectrum, which is further confirmed in the ARPES data (inset). **d** Enlarged topographic image of $[Gd_2C]^{2+}$ surface. $V_{bias} = 25$ mV and $I_t = 100$ pA. **e** $dI/dV$ map at $V_{bias} = 25$ mV. $V_{mod} = 10$ mV. **f** $dI/dV$ map at $V_{bias} = -100$ mV. $V_{mod} = 10$ mV.

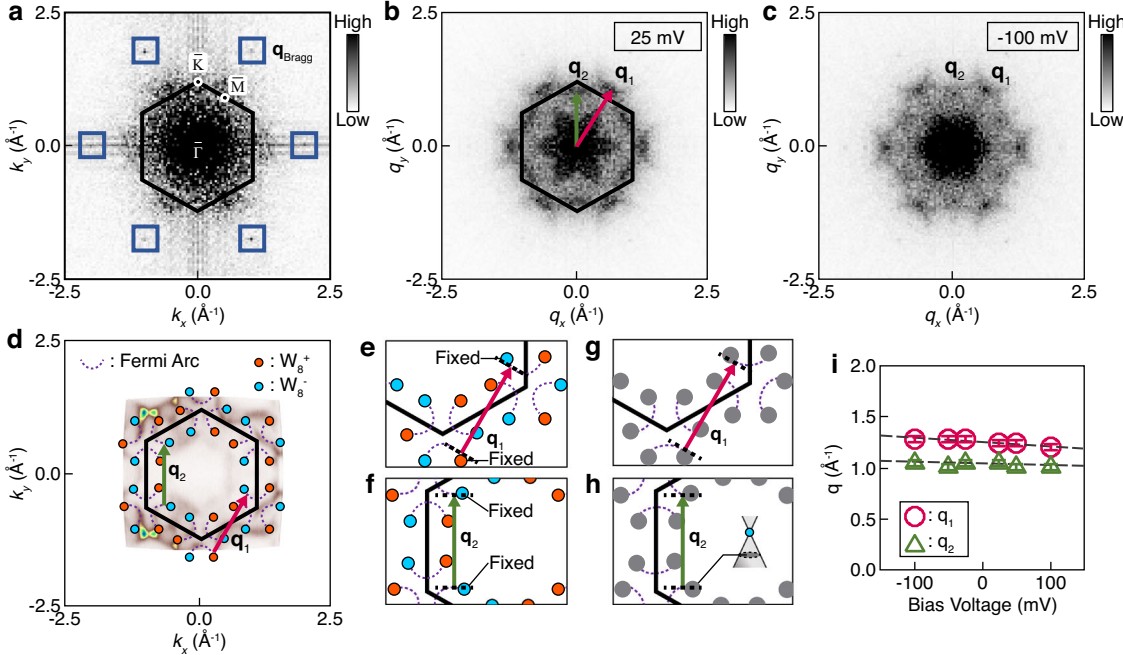

**Fig. 6 | STM analysis on Fermi-arc states of $[Gd_2C]^{2+}\cdot2e^-$. a** Fourier transformed the image of the topographic image (Fig. 5d). The blue boxes indicate the Bragg peaks. **b**, **c** Fourier transformed images of the $dI/dV$ maps (Fig. 5e, f). Two distinct QPI modulation vectors are identified as $\mathbf{q}_1$ (red arrow) and $\mathbf{q}_2$ (green arrow). **d** Schematics illustrating Fermi-arc states, overlaid with ARPES data for clarity. The $\mathbf{q}_1$ and $\mathbf{q}_2$ vectors joining the Fermi-arc states are assigned by considering their sizes and directions. **e**, **f** Nesting conditions for the $\mathbf{q}_1$ and $\mathbf{q}_2$ vectors. **g**, **h** Nesting conditions for $\mathbf{q}_1$ and $\mathbf{q}_2$ at a lowered bias voltage. The dispersion of Fermi-arc states is minimal near the Weyl cone, leading the $\mathbf{q}_1$ and $\mathbf{q}_2$ vectors to remain unaltered significantly. **i** Dispersion of $q_1$ (red circles) and $q_2$ (green triangles) by the bias voltage. The dashed lines are given for eye-guides. The error bar represents the momentum uncertainty resulting from the resolution of $dI/dV$ maps.

Three-fold symmetry as a rhombohedral structure in the $\phi$-scan (azimuthal scan) X-ray diffraction pattern and the negligible impurity level (<1 ppm) in the inductively coupled plasma spectroscopy results were obtained, guaranteeing the high quality of the single crystals.

## ARPES measurements

ARPES measurements were performed at beamline 4.0.3 (MERLIN) of Advanced Light Source (ALS), Lawrence Berkeley National Laboratory (LBNL). Samples were prepared in a glove box filled with Ar gas to prevent contamination from water and oxygen. $[Gd_2C]^{2+}\cdot2e^-$ single

crystals were attached to the ARPES sample holders and covered by ceramic top posts for in-situ cleaving in the APRES chamber. Single crystal samples and top posts were glued with Muromac H-220 conductive silver epoxy. After curing the silver epoxy, the samples were immediately transferred to the UHV chamber for ARPES measurement. Prepared single crystal samples were in-situ cleaved at 10 K under ultra-high vacuum better than $4 \times 10^{-11}$ Torr to obtain clean surfaces. All measurements were performed at a temperature of 10 K. Spectra were acquired with a Scienta R8000 analyzer. Photon energies for the measurements are given in each figure.

### STM measurements
STM measurements were performed using a home-built cryogenic STM working at 4.3 K. The $[Gd_2C]^{2+} \cdot 2e^-$ single crystal was glued on the sample holder using a conducting epoxy (EPO-TEK, H20E) in a glove box filled with Ar gas. A cleaving post was subsequently glued on the sample using a non-conducting epoxy (EPO-TEK, H74F). The sample was then transferred to the UHV chamber and cleaved below 20 K under the vacuum pressure better than $5 \times 10^{-11}$ Torr. All the measurements were conducted at 4.3 K. The $dI/dV$ spectra and $dI/dV$ maps were acquired using a standard lock-in technique (Signal Recovery, Model 7265) with a modulation frequency $f = 716.3$ Hz.

### DFT calculations
DFT calculations were performed using the generalized gradient approximation with the Perdew–Burke–Ernzerhof (PBE) functional[31] and the projector augmented plane-wave method[32] implemented in the Vienna Ab initio Simulation Program code[33]. The $4f$, $5s$, $5p$, $5d$, and $6s$ electrons of Gd and the $2s$ and $2p$ electrons of C were used as valence electrons. The plane-wave-basis cut-off energy was set to 600 eV. We have chosen a sufficiently thick slab model of $[Gd_2C]^{2+} \cdot 2e^-$ consisting of 18 atoms and a vacuum layer of 20 Å along the $c$ axis was used for the surface calculation. The middle layer of the slab was fixed in bulk position. The geometrical relaxation of the slab was performed using $8 \times 8 \times 1$ $k$-point meshes until the Hellmann–Feynman forces were less than $10^{-5}$ and $10^{-3}$ eV/Å, respectively. The empty spheres with a Wigner–Seitz radius of 1.25 Å were placed both on the surface and at the interlayer space to obtain the projected density of states of the surface and interlayer positions, respectively.

## Data availability
All data supporting the findings of this work are included in the main text and supplementary information. These are available from the corresponding authors upon request.

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

## Acknowledgements

C.-y.L, J.C., G.L., and Y.K. were supported by the National Research Foundation of Korea (NRF) grant funded by the Korea government (MSIT) (2022M3H4A1A01010832), and Samsung Science and Technology Foundation under project number SSTF-BA2101-04. S.K. was supported by the National Research Foundation of Korea (NRF) funded by the Ministry of Education (Grants No. 2021R1A6A1A10044950, No. RS-2023-00285390). M.-S.K., S.Y.S., and J.S. were supported by the National Research Foundation of Korea (NRF) grant funded by the Ministry of Science and ICT (RS-2023-00209704). D.T. and S.-G.K. utilized computer time allocation provided by the High Performance Computing Collaboratory (HPC2) at Mississippi State University.

## Author contributions

Y.K. and S.W.K. designed and directed the project. D.C.L. and S.W.K. synthesized the samples. C.-y.L., S.K., Y.L., J.C., G.L., J.D.D., and Y.K. performed the ARPES measurements. D.T., S-.G.K., Y.K., and S.W.K. delivered the DFT calculations. M.-S.K and S.Y.S. conducted the STM experiments under the supervision of J.S. Y.K., J.S., and S.W.K. co-wrote the manuscript with support from all authors.

## Competing interests

The authors declare no competing interests.
