## [Peer Review file · Nature Communications]

REVIEWER COMMENTS

Reviewer #1 (Remarks to the Author):

Chan-young Lim and coworkers present a study in which they apply several methods to uncover the electronic structure of a fascinating material. The methods appear to have been applied skillfully and seem to corroborate the theoretically predicted nature of the compound. There are two areas I think the authors should address:

The crystal termination presented in Fig. 1a is polar. In this case the charges in the ionic layers will have to differ from the bulk values to avoid divergence of the electrostatic surface energy. The layered electrides might be uniquely adaptable in this respect, because the two dimensional electron pool can presumably accommodate exactly the required compensating charge. It is therefore that I think the surface modification process to remove the outer electron layer deserves more attention. Presumably the low sample cleaving temperature of 10 K is insufficient to allow for any structural changes to respond to the creation of a (polar) surface.

The authors observe that subsequent heating to 80 K apparently causes the outer electron layer to vanish, but this must be accompanied by a charge redistribution that satisfies the electrostatic conditions in another way. The uncovered Gd₂C layer is therefore likely to differ from its bulk counterparts. I would encourage the authors to address this issue in more detail.

The photon energy dependent ARPES results in Fig. 3d appear to corroborate the interpretation in terms of a three dimensional Weyl cone. Variation of the matrix element with photon energy can however be deceiving when trying to interpret results merely on the basis of intensity. It is stated that k_z approaches a zone boundary at the indicated photon energy. This statement could be made more convincing by demonstrating that the effect also occurs in a different photon energy range corresponding to the same k_z values.

Reviewer #2 (Remarks to the Author):

This paper is a study of the topological surface states of the ferromagnetic Weyl semimetal Gd₂C, a two-dimensional electride. The bulk is a Weyl semimetal with broken time-reversal symmetry, and due to the topological nature of the bulk, a Fermi arc appears at the surface connecting the Weyl points. In this paper, the surface states are investigated by ARPES, STM, and DFT, and their spatial locations are also identified.

The authors examine topological electrides exhibiting magnetic Weyl semimetallic phases from multiple and complementary angles. The results are also interesting because of the relevance of both topological and electride aspects. However, the interpretation of the results and the basis of the results are still open for consideration. Also there are several prior studies on Gd₂C, and it seems that this paper does not contain new findings worthy of publication in Nature Communications. Therefore, the content of the current manuscript is not suitable for publication in Nature Communications. Below are my comments.

Comment 1

Much of the analysis of Gd₂C as a Weyl semimetallic phase is explained in Phys. Rev. Lett. 125,187203 (2020) and Nature Communications 11, 1526 (2020) etc.. Where is the novelty in this paper as an understanding of the topological properties of Gd₂C?

Comment 2

It is difficult to understand the explicit meaning of "Bulk Weyl state", "Floating electron state", and "Fermi arc state" in Fig1a.

Comment 3

Although the authors argue that "It is noteworthy that the bands with dominant contribution from the electrons in the topmost Gd atoms are distinct from previously observed floating bands and bulk IAE bands, implying the existence of additional surface states at topmost atomic layer.", both Fig.1b and Fig.1c are similar in the location of the strong intensity and the spin orientation. This suggests that they are hybridized and is not a reason for additional topological bands on the surface.

Comment 4

The floating state in Gd₂C has a small work function and the shape of the wavefunction is s-orbital-like and less localized. Is there any possibility that the STM probe itself can affect the floating state during the measurement?

Comment 5

In the ARPES result (Fig1e), the rotational symmetry C₃ seems to be broken at the surface. Is this a problem in accuracy? Or is it some other intrinsic effect, such as C₃ symmetry breaking due to spin orientation in the xy-plane?

Comment 6

The authors remove the floating state by heating and perform STM measurements again. Is the bulk region unchanged by heating? If the floating state on the surface disappears, the bulk may also be affected.

Comment 7

The surface after heating is no longer a separate structure from the surface before heating. The motivation to see the surface state of the Gd origin beneath the pre-heating floating state is understandable, but would not be apparent from the post-heating experiment. The authors may have used the consistency with ARPES and DFT as the validity of the STM after heating. However, the STM result in Fig.4 is unclear, and the Fermi arc in Fig. 4j is not the experimental result.

Comment 8

The Fermi arc is a surface state, but the Weyl point, to which the Fermi arc connects, is a bulk state; the Fermi arc in the very vicinity of the Weyl point also does not have much of a surface contribution. Where is the justification for bringing up the Weyl point, a bulk state, in a discussion of q_1 and q_2 at the surface?

Reviewer #3 (Remarks to the Author):

The manuscript studies an two-dimensional (2D) electrides use ARPES and STM, and compares the floating electron states and their topological states. The material and the results are very interesting. But the manuscript cannot be published because of the main concerns below.

In order to remove the floating electrons, the sample was heated to 80 K and cooled to 4 K after an hour. As the surface and surface states are very sensitive to the environment (for example, residual gas during heating), this procedure might have changed the surface dramatically. So it is not safe to claim the difference between the two STM data in Fig. 2 and Fig. 4 is only “remove the floating electrons”. Actually, the Fig.4 images are rougher than Fig.2 images, with more “defects” like features on the surface.

If 80 K heating can remove the floating electrons, how does a 80 K cleavage fresh surface look like?

How does the heating procedure change the ARPES results?

REVIEWER COMMENTS

Reviewer #1 (Remarks to the Author):

Chan-young Lim and coworkers present a study in which they apply several methods to uncover the electronic structure of a fascinating material. The methods appear to have been applied skillfully and seem to corroborate the theoretically predicted nature of the compound. There are two areas I think the authors should address:

The crystal termination presented in Fig. 1a is polar. In this case the charges in the ionic layers will have to differ from the bulk values to avoid divergence of the electrostatic surface energy. The layered electrides might be uniquely adaptable in this respect, because the two dimensional electron pool can presumably accommodate exactly the required compensating charge. It is therefore that I think the surface modification process to remove the outer electron layer deserves more attention. Presumably the low sample cleaving temperature of 10 K is insufficient to allow for any structural changes to respond to the creation of a (polar) surface. The authors observe that subsequent heating to 80 K apparently causes the outer electron layer to vanish, but this must be accompanied by a charge redistribution that satisfies the electrostatic conditions in another way. The uncovered Gd₂C layer is therefore likely to differ from its bulk counterparts. I would encourage the authors to address this issue in more detail.

Author Reply: We appreciate this insightful comment. We agree with the reviewer that the system's charge distribution should be adjusted to meet the new electrostatic conditions after removing the floating electrons. In [Gd₂C]²⁺·2e⁻, interstitial anionic electrons (IAEs) are squeezed between [Gd₂C]²⁺ layers, and the electron densities in these layers change to screen the surface charges. Since STM cannot directly measure the changes in electron density in IAEs, we used ARPES to inquire about the band topology of a Gd₂C sample subjected to the same heating process as in STM. The ARPES results clearly indicate the presence of Fermi arc states on this heated sample, while showing the absence of the floating electron state (Fig. R1). This absence confirms that the sample is heated, consistent with the STM results.

The ARPES data show a slight Fermi level shift in bulk bands due to the charge redistribution. However, the presence of the topological surface state demonstrates that the charge redistribution does not alter the band topology of the system. This is understandable, as the density changes in IAEs are at most a factor of 2, even for the topmost IAE. In addition, the Gd lattice structure is seen in the STM data, indicating that the Gd surface remains intact after the heating.

To compile the reviewer's comments, we have added the following discussions in the revised manuscript.

“Given that no floating electrons are present on the surface, the electrons under the surface must be accommodated to satisfy the electrostatic conditions. Within $[\text{Gd}_2\text{C}]^{2+} \cdot 2\text{e}^-$, IAEs are squeezed between $[\text{Gd}_2\text{C}]^{2+}$ layers, and their electron densities will change to screen the surface charges. Nevertheless, the ARPES data measured on Gd_2C , which underwent the same heating process as in STM, clearly show the presence of Fermi arc states on the surface (Extended Data 3), although there is a slight Fermi level shift. This demonstrates that the charge redistribution does not alter the bulk topology of the system.” (Page 7)

“In the FT of the STM image (Fig. 4g), the lattice peaks (q_{Bragg}) of the Gd layer are identified, helping to define the BZ. The presence of the Gd lattice peaks indicates that there is no surface modification upon the removal of the floating electrons.” (Page 7)

Figure R1. Band dispersions along $\text{M}-\Gamma-\text{M}$ high symmetry direction taken from the fresh cleaved surface (a) and after the heating cycle (b). While the parabolic band of floating electrons disappears after heating, the bulk band dispersion is almost intact, with a slight shift of the Fermi level toward higher binding energy.

Thanks to the reviewer, we recognized that we need to discuss the intact bulk state after the heating cycle, not just provide an intact Fermi arc. Therefore, we revised the Extended Data Figure 3 as below.

The photon energy dependent ARPES results in Fig. 3d appear to corroborate the interpretation in terms of a three dimensional Weyl cone. Variation of the matrix element with photon energy can however be deceiving when trying to interpret results merely on the basis of intensity. It is stated that k_z approaches a zone boundary at the indicated photon energy. This statement could be made more convincing by demonstrating that the effect also occurs in a different photon energy range corresponding to the same k_z values.

Author Reply: We thank the reviewer for this comment. Despite our attempts, the suggested experiments were not feasible for practical reasons. Other photon energies for scanning over $k_z = \pi$ positions are about 110 eV and 40 eV. The photon energy 40 eV is not favorable because setting a large energy window to cover a given k_z range would introduce a complicated photon energy-dependent matrix element effect. At 110 eV, the intensity of the Weyl bands was unfortunately too weak in our measurement.

Instead, we discuss the uneven intensity between the two intersecting bands, where one band exhibits apparent intensity compared to the other. This band-dependent effect of matrix element is primarily caused by a combination of the spatial symmetry of the corresponding wave function and the experimental geometry, including the polarization of light and direction of light incidence. However, this effect does not seem to significantly change within the small range of photon energy. In fact, the overall tendency consistently appears within the photon energy range used in Fig. 3d.

In recognition of the reviewer's concern, we made modifications to the revised manuscript to provide a detailed explanation for determining the Weyl point.

"In both plots, the cone-like dispersions are clearly shown, although the intensity of one band is dominating among the two intersecting bands. The uneven intensity observed between two intersecting bands is frequently observed in cases of Dirac/Weyl crossings, resulting from the differing symmetries of the corresponding wave functions of each band concerning the experimental geometry of light polarization and incident light direction. The smooth intensity profile, lacking a true dip along the dominant band, can be attributed to the band crossing occurring without hybridization with the minor band. Hybridization typically encodes the character of the minor band into the dominant band. By considering the intensity profile of the dominant band, and by tracing the dispersion of the minor band against the dominant band, the intersecting point for $W_{13/14}^+$ was determined to be located at $k_z = \pi$, as predicted." (Pages 5-6)

Reviewer #2 (Remarks to the Author):

This paper is a study of the topological surface states of the ferromagnetic Weyl semimetal Gd₂C, a two-dimensional electride. The bulk is a Weyl semimetal with broken time-reversal symmetry, and due to the topological nature of the bulk, a Fermi arc appears at the surface connecting the Weyl points. In this paper, the surface states are investigated by ARPES, STM, and DFT, and their spatial locations are also identified.

The authors examine topological electrides exhibiting magnetic Weyl semimetallic phases from multiple and complementary angles. The results are also interesting because of the relevance of both topological and electride aspects. However, the interpretation of the results and the basis of the results are still open for consideration. Also there are several prior studies on Gd₂C, and it seems that this paper does not contain new findings worthy of publication in Nature Communications. Therefore, the content of the current manuscript is not suitable for publication in Nature Communications. Below are my comments.

Comment 1

Much of the analysis of Gd₂C as a Weyl semimetallic phase is explained in Phys. Rev. Lett. 125,187203 (2020) and Nature Communications 11, 1526 (2020) etc.. Where is the novelty in this paper as an understanding of the topological properties of Gd₂C?

Author Reply: We thank the reviewer for his/her effort in reviewing our manuscript. The paper [Phys. Rev. Lett. 125, 187203 (2020)] theoretically discusses the Weyl state. Our work demonstrates the presence of Weyl state and Fermi arc state experimentally. Until our work, the topological aspects of electride materials have not been confirmed by experiments, which enforces the novelty of our report and should be acknowledged.

Moreover, we show that floating electrons on the surface can be removed, which provides counter evidence that the electron layer can exist independently, separated from the surface. This is an experimental remark addressing any doubts about the floating electron layer.

Combining two findings, our work realizes the unique stacking of two different surface states; floating electron state originating from the electrostatic characteristic of electride and Fermi arc state based on the symmetry of electride. This work not only completes the exotic states possible for electride system, but also highlights exciting avenues to unexplored territory of complex quantum phases with the combination of two distinct surface states. We believe these implications of our work hold novelty merits to the readers of Nature Communications.

The paper [Nature Communications 11, 1526 (2020)], which is reported by one of our corresponding authors, does not deal with the topological phase in Gd₂C but focuses on the material properties of Gd₂C, mainly on the magnetism.

Comment 2

It is difficult to understand the explicit meaning of "Bulk Weyl state", "Floating electron state", and "Fermi arc state" in Fig1a.

Author Reply: We thank the reviewer for this comment. These terminologies were used to represent our main discovery graphically. In the revised manuscript, we explain their meanings in the figure caption as below.

“Schematic of the crystal (left panel) and electronic (right panel) structures. In the left panel, the green and black spheres represent Gd and C atoms, respectively, forming $[Gd_2C]^{2+}$ layers. Blue blobs between the $[Gd_2C]^{2+}$ layers denote bulk interstitial anionic electrons (IAEs). The floating electrons are depicted atop the crystal structure. The right panel illustrates the electronic structure corresponding to the crystal structure. The bulk of the crystal exhibits a non-trivial band topology, hosting the bulk Weyl state. The Fermi arc state is observed on the $[Gd_2C]^{2+}$ surface, while the floating electron state reveals a circular Fermi surface.” (Page 11)

Comment 3

Although the authors argue that “It is noteworthy that the bands with dominant contribution from the electrons in the topmost Gd atoms are distinct from previously observed floating bands and bulk IAE bands, implying the existence of additional surface states at topmost atomic layer.”, both Fig.1b and Fig.1c are similar in the location of the strong intensity and the spin orientation. This suggests that they are hybridized and is not a reason for additional topological bands on the surface.

Author Reply: As the reviewer pointed out, the atoms in $[Gd_2C]^{2+} \cdot 2e^-$ are hybridized with the floating electrons and bulk IAE. This is the mechanism that $[Gd_2C]^{2+} \cdot 2e^-$ acquires a non-trivial band topology. Therefore, we cannot fully separate the Fermi arc states from the intertwined bands in the system. Nevertheless, the primary contributions of Gd atoms, floating electrons, and bulk IAE to the bands can still differ, as the contribution weight is represented by the size of the circles in Fig. 1b-1d. In Fig. 1c, it is apparent that the Fermi arc state is dominantly contributed by the topmost Gd atomic orbitals. To compile the reviewer’s comment, we marked an arrow in Fig. 1c to indicate the position of the Fermi arc state we are referring.

Comment 4

The floating state in Gd₂C has a small work function and the shape of the wavefunction is s-orbital-like and less localized. Is there any possibility that the STM

probe itself can affect the floating state during the measurement?

Author Reply: In principle, the electric field from the STM tip can affect the measurement of materials. However, in metallic materials like $[\text{Gd}_2\text{C}]^{2+}\cdot 2\text{e}^-$, the electric field is efficiently screened by the free carriers. The extracted k vectors from the QPI patterns closely match the ARPES data (Fig. 2 in the main text), indicating that the influence of the STM tip during the measurement should be minimal.

Comment 5

In the ARPES result (Fig1e), the rotational symmetry C_3 seems to be broken at the surface. Is this a problem in accuracy? Or is it some other intrinsic effect, such as C_3 symmetry breaking due to spin orientation in the xy -plane?

Author Reply: We would like to clarify that the Fermi surface given in Fig. 1e is symmetrized with respect to the $G - K'$ direction for better visualization, as clarified in the caption. In APRES measurements, the matrix element effect, which is influenced by the polarization and incident directions of the probing light, commonly alters the overall intensity profile, distorting the symmetric pattern of the photoelectrons in the excited state that we measure. Still, some hints of C_3 symmetry can be seen from the different intensity patterns around M and M' points in Fig. 1e.

Comment 6

The authors remove the floating state by heating and perform STM measurements again. Is the bulk region unchanged by heating? If the floating state on the surface disappears, the bulk may also be affected.

Author Reply: We appreciate the reviewer's comment. We understand the reviewer's concern is about the crystalline stabilities of Gd_2C after removing the floating electrons. However, we want to emphasize that Gd_2C is not destroyed by heating the sample. First, The Gd lattice peaks are clearly visible in the Fourier transform (FT) image of the Gd_2C surface (Fig. 4g in the main text, and also Fig. R2a and b), indicating that the surface maintains its crystallinity. Second, using ARPES, we examined the band structure of the Gd_2C sample that underwent the same heating process as in STM. The data show that while the floating electron state is absent, the features of other bulk states and Fermi arc states remain unchanged (Fig. R2c). This demonstrates that the heating process removes the floating electrons but does not alter the band dispersion of the Gd_2C , preserving its band topology.

Figure R2. (a) STM image of heated Gd_2C sample. Despite residual impurities, the Gd surface is largely open for STM measurements. (b) Fourier transformed image of (a). The green circles indicate the Gd lattice peaks. The size of the peaks is $\sim 2 \text{ \AA}^{-1}$, agreeing with the ARPES data. (c) ARPES data measured on the heated sample. The dotted circles indicate the Fermi arc states. The floating electron states at the Γ point do not exist in the heated sample.

Comment 7

The surface after heating is no longer a separate structure from the surface before heating. The motivation to see the surface state of the Gd origin beneath the pre-heating floating state is understandable, but would not be apparent from the post-heating experiment. The authors may have used the consistency with ARPES and DFT as the validity of the STM after heating. However, the STM result in Fig.4 is unclear, and the Fermi arc in Fig. 4j is not the experimental result.

Author Reply: Fermi arc states in Fig. 4j are depicted from ARPES data, as explained in the main text. To compile the reviewer's comment, we overlap the schematic with the ARPES data in the revised manuscript. There are some impurities like crystalline defects in Fig. 4d, which cannot be avoidable in practical crystals. Nevertheless, the STM data is clear enough to show the QPIs resulting from the Fermi arc states, and it is apparent that the presence of the electron residues does not disrupt the QPI patterns.

Comment 8

The Fermi arc is a surface state, but the Weyl point, to which the Fermi arc connects, is a bulk state; the Fermi arc in the very vicinity of the Weyl point also does not have much of a surface contribution. Where is the justification for bringing up the Weyl point, a bulk state, in a discussion of q_1 and q_2 at the surface?

Author Reply: We agree with the reviewer that the Fermi arc near the Weyl points might not have much of a surface contribution. However, it should also be noted that

the Fermi arc states near the Weyl points provide the strongest Fermi surface nesting. Therefore, the resulting QPI intensities arise from balancing these two factors. In our experiments, we observed apparent modulation vectors of QPI. By comparing their sizes and directions with the Fermi arc states measured by ARPES, and considering their less-dispersive nature, we concluded that the q_1 and q_2 vectors connect to the Fermi arc states in the vicinity of the Weyl point, but not necessarily exactly at the Weyl points.

Reviewer #3 (Remarks to the Author):

The manuscript studies a two-dimensional (2D) electride using ARPES and STM, and compares the floating electron states and their topological states. The material and the results are very interesting. But the manuscript cannot be published because of the main concerns below.

In order to remove the floating electrons, the sample was heated to 80 K and cooled to 4 K after an hour. As the surface and surface states are very sensitive to the environment (for example, residual gas during heating), this procedure might have changed the surface dramatically. So it is not safe to claim the difference between the two STM data in Fig. 2 and Fig. 4 is only “remove the floating electrons”. Actually, the Fig. 4 images are rougher than Fig. 2 images, with more “defects” like features on the surface.

Author Reply: We thank the reviewer for his/her efforts in reviewing our manuscript. The reviewer's concern is about the validity of the surface after the heating process. We acknowledge this concern, which is precisely why we conducted ARPES experiments in addition to the STM study. Please see that this work is a complementary study between STM and ARPES. The validity of the work cannot be judged by either method alone.

The ARPES measurements on the Gd_2C sample, which underwent the same heating process as in the STM study, clearly show the presence of Fermi arc states. The floating electron state is missing here, assuring the sample is heated. The absence of the floating electron state perfectly agrees with the STM observations. The bulk band dispersion is nearly intact, with a slight shift in the Fermi level (Fig. R3). Moreover, ARPES relies on a beam penetration depth of a few Å. If the surface had been destroyed or significantly changed during the heating process, we would not have observed the Fermi arc states in the ARPES measurements.

Given that the ARPES confirms the presence of Fermi arc states, we discuss the impurities on the surface. In the STM images, there are impurities on the Gd surface (Fig. R4a). Some are crystalline defects, and others could have been introduced during the heating process. However, the Gd surface is still open wide enough to

show QPI patterns resulting from the Fermi arc states. Importantly, the Fermi arc states originate from a non-trivial bulk topology. Therefore, they should not be sensitive to surface impurities if the bulk topology is preserved. In fact, the dI/dV maps show that the impurities play the role of scattering centers but do not disrupt the QPI patterns (Figs. R4b and R4c).

Given combined evidence from ARPES and STM, we hope the reviewer agrees that the Gd surface in the heated sample is valid for studying the Fermi arc states.

Figure R3. Band dispersions along $M-\Gamma-M$ high symmetry direction taken from the fresh cleaved surface (a) and after the heating cycle (b). While the parabolic band of floating electrons disappears after heating, the bulk band dispersion is almost intact, with a slight shift of the Fermi level toward higher binding energy.

Figure R4. (a) STM topography of Gd surface measured at $V_{\text{bias}} = 50$ mV. (b) dI/dV map at $V_{\text{bias}} = 50$ mV. (c) Fourier transformed image of (b). There are residual impurities on the surface, but they do not hinder the observation of Fermi arc states.

If 80 K heating can remove the floating electrons, how does a 80 K cleavage fresh surface look like? How does the heating procedure change the ARPES results?

Author Reply: We thank the reviewer for this intuitive question. In response, we examined the $[\text{Gd}_2\text{C}]^{2+} \cdot 2e^-$ sample cleaved at 80 K using STM. The sample shows a qualitatively similar surface to the heated sample, implying that the surface impurities

are not introduced in association with residual gases (Fig. R5).

Figure R5. Gd_2C sample cleaved at 80 K. The image is measured at $V_{\text{bias}} = 50$ mV. The measurement temperature is 4 K. Compared to Fig. R4, the residual impurities appear approximately the same on the surface.

REVIEWERS' COMMENTS

Reviewer #1 (Remarks to the Author):

All reviewers expressed concerns about the possibility of changes induced by the heating procedure to remove the outer floating electron layer. I think that the authors have dealt with this topic satisfactorily: The more careful discussion of this matter in the revised manuscript points out the small effect seen in the bulk band structure. In addition, the investigation of a sample cleaved at higher temperature serves as a good test to their interpretation.

With these improvements I can endorse the work for publication.

Reviewer #2 (Remarks to the Author):

This paper is a study of the topological surface states of the ferromagnetic Weyl semimetal Gd₂C, a two-dimensional electride. The bulk is a Weyl semimetal with broken time-reversal symmetry, and due to the topological nature of the bulk, a Fermi arc appears at the surface connecting the Weyl points. In this paper, the surface states are investigated by ARPES, STM, and DFT, and their spatial locations are also identified.

The authors have responded appropriately to all questions from referees.

This paper is suitable for publication in Nature Communications.

Reviewer #3 (Remarks to the Author):

The authors addressed my comments and concerns. I agree this manuscript is ready for publication.

REVIEWER COMMENTS

Reviewer #1 (Remarks to the Author):

All reviewers expressed concerns about the possibility of changes induced by the heating procedure to remove the outer floating electron layer. I think that the authors have dealt with this topic satisfactorily: The more careful discussion of this matter in the revised manuscript points out the small effect seen in the bulk band structure. In addition, the investigation of a sample cleaved at higher temperature serves as a good test to their interpretation.

With these improvements I can endorse the work for publication.

Author Reply: We would like to thank the reviewer for acknowledging our effort to revise the manuscript. With fruitful comments from the reviewer, we could improve our manuscript.

Reviewer #2 (Remarks to the Author):

This paper is a study of the topological surface states of the ferromagnetic Weyl semimetal Gd₂C, a two-dimensional electride. The bulk is a Weyl semimetal with broken time-reversal symmetry, and due to the topological nature of the bulk, a Fermi arc appears at the surface connecting the Weyl points. In this paper, the surface states are investigated by ARPES, STM, and DFT, and their spatial locations are also identified.

The authors have responded appropriately to all questions from referees.

This paper is suitable for publication in Nature Communications.

Author Reply: We would like to appreciate the reviewer for recognizing the value of our work and recommending it for the publication in Nature Communications.

Reviewer #3 (Remarks to the Author):

The authors addressed my comments and concerns. I agree this manuscript is ready for publication.

Author Reply: We would like to thank the reviewer for recommending the publication of our manuscript in Nature Communications.